# Dynamic Alteration Profile and New Role of RNA m6A Methylation in Replicative and H_2_O_2_-Induced Premature Senescence of Human Embryonic Lung Fibroblasts

**DOI:** 10.3390/ijms23169271

**Published:** 2022-08-17

**Authors:** Fan Wu, Luyun Zhang, Caiyun Lai, Xinyue Peng, Susu Yu, Cheng Zhou, Bo Zhang, Wenjuan Zhang

**Affiliations:** 1Food Safety and Health Research Center, School of Public Health, Southern Medical University, Guangzhou 510515, China; 2Department of Public Health and Preventive Medicine, School of Medicine, Jinan University, Guangzhou 510632, China

**Keywords:** oxidative stress, premature senescence, m6A, RNA methylation

## Abstract

N6-methyladenosine (m6A) methylation is one of the most common RNA modifications, regulating RNA fate at the posttranscriptional level, and is closely related to cellular senescence. Both models of replicative and premature senescence induced by hydrogen peroxide (H_2_O_2_) were used to detect m6A regulation during the senescence of human embryonic lung fibroblasts (HEFs). The ROS level accumulated gradually with senescence, leading to normal replicative senescence. H_2_O_2_-treated cells had dramatically increased ROS level, inducing the onset of acute premature senescence. Compared with replicative senescence, ROS changed the expression profiles for m6A-related enzymes and binding proteins, including higher levels of METTL3, METTL14, WTAP, KIAA1429, and FTO, and lower levels of METTL16, ALKBH5, YTHDC1, and YTHDF1/2/3 in the premature senescence persistence group, respectively. Meanwhile, senescent cells decreased total m6A content and RNA methylation enzymes activity, regardless of replicative or premature senescence. Moreover, specific m6A methylation levels regulated the expression of SIRT3, IRS2, and E2F3 between replicative and premature senescence separately. Taken together, differential m6A epitranscription microenvironment and the targeted genes can be used as epigenetic biomarkers to cell senescence and the related diseases, offering new clues for the prevention and intervention of cellular senescence.

## 1. Introduction

Senescence is a process in which the functions of various tissues and organs of the body show degenerative changes with age [1], reducing the ability of dynamic balance and increasing the chances of illness and death. It is also a risk factor for age-related diseases, such as hypertension, type 2 diabetes, atherosclerosis, and neurodegenerative diseases, etc. Nowadays, antiaging and healthy aging have great significance to longevity and health. Senescence generally includes replicative senescence and premature senescence induced by extrinsic environmental or occupational factors. Cellular replicative senescence was first discovered by Hayflick, in which with the increase in age, human diploid cells cultured in vitro became larger, metabolism disorders occurred, intracellular particles increased, and cell proliferation ability gradually stopped [2,3]. When human embryonic lung fibroblasts meet with exogenous H_2_O_2_, they are susceptible to oxidative stress and eventually own senescent characteristics [4,5], which provide a useful model for studying the potential mechanisms and markers associated with aging.

Redox signals play important roles in cellular life cycles, maintaining physiological homeostasis, the imbalance of which leads to an increase in active oxygen or a decrease in clearance, impairing the functions of biomolecules and causing cell senescence [6]. H_2_O_2_, as a main redox product in the body, has important roles in redox sensing, signal transduction, and redox regulation [7]. H_2_O_2_ exceeding physiological concentrations (>100 nmol/L) can cause oxidative damages to biomolecules, which diffuses through cells and tissues, and triggers cellular effects such as changing cell shape, proliferating, and recruiting immune cells [8,9]. The destructive effects of oxidants are consistent with the characteristics of cellular replicative senescence, including mitochondrial dysfunction, protein denaturation and aggregation, cell membrane and intercellular communication damages, regenerative cells loss, and genome instability. The H_2_O_2_-induced oxidative stress model is used to explore the mechanism of oxidative stress causing damages to the body [10]. Ginsenoside Rb1 reduces H_2_O_2_-induced senescence of human umbilical vein endothelial cells by stimulating the sirtuin-1/AMP-activated protein kinase pathway [11]. The internal mechanisms of ROS inducing senescence, however, require further clarification, with the epitranscriptome regulation of premature senescence being still in its infancy.

RNA modification affects the development of many eukaryotic organisms and m6A methylation is the most extensive posttranscriptional modification, involving RNA nucleation, splicing, stability, etc. [12,13]. The m6A methylation accounts for 0.1–0.4% of adenosine in whole-cell RNA and approximately 50% of all methylated ribonucleotides [14], which is associated with many physiological and pathological events related with aging, such as obesity, immunoregulation, and tumors [15]. The biological functions of RNA m6A are mainly regulated by RNA methyltransferases, demethylases, and binding proteins jointly. The m6A binds to mRNA through RNA methyltransferase, which can be cancelled by demethylase FTO and ALKBH5 and the process of which is dynamic and reversible. The m6A modification can affect the binding of methylated mRNA to proteins and regulate the metabolism of those transcripts directly or indirectly [16].

Cellular-senescence-related genes have formed a complex network, possibly modified by m6A RNA methylation, which may also interact with oxidative stress [13,17,18,19]. In this study, we will detect the effects of ROS, the dynamic alterations of m6A RNA methylation, and the network of targeted genes during cellular senescence. These characteristics will provide profile differences between cellular replicative and premature senescence, and potential intervention clues for age-related diseases.

## 2. Results

### 2.1. Accumulating Oxidative Stress Induces Cellular Senescence

Based on the cellular premature senescence model induced by 400 μmol/L H_2_O_2_ in our previous study [20], we first verified the senescent status by cell morphology (Figure 1A) and the results of SA-β-gal staining (Figure 1C). Following acute treatment with H_2_O_2_, the cells gradually exhibited the typical phenotype of premature senescence, involving enlarged nuclei and increased cytoplasmic granules. The expression of SA-β-gal increased in senescent cells. The rate of blue staining cells in the 49PDL group and the PSp group was 42.2 and 37.9 times the rate of those in the 22PDL group, respectively.

Then, the total cellular ROS level was detected to search the reasons for senescence occurrence. Shown in Figure 1B, the total cellular ROS increased during cellular replicative senescence. To the premature senescence, the ROS level increased sharply following acute H_2_O_2_ treatment in the PSi group; then, it decreased gradually until cells entered the stable status of premature senescence constantly. Compared with that of the 22PDL group, ROS content of the 49PDL group and the PSi group increased by 3.4 times and 2.5 times, respectively, and the difference was statistically significant (*p* < 0.05). However, it decreased to the same level in the PSp group as in the 22PDL group. The ROS content of the PSi group was higher than that of the PSp group. In the PSi group with the acute H_2_O_2_ treatment, cells accumulated excessive ROS in a short period. After the following continuous culture for 7 days with normal medium, cells eliminated a certain level of ROS, but they showed persistent and irreversible senescence phenotype with the higher content of SA-β-gal in the PSp group than that in the PSi group (*p* < 0.05). Therefore, the accumulation of intracellular ROS was accompanied by oxidative damages, promoting cellular replicative senescence. After H_2_O_2_ treatment, the acute oxidative damage was caused in the short period. Although the ROS content decreased gradually later, irreparable oxidative damages still existed continuously and cells entered irreversibly premature senescence persistence status. Excessive ROS was the direct external cause to cellular premature senescence.

### 2.2. Global m6A Profile of Senescent Cells Is Modulated by Oxidative Stress

To assess the global m6A RNA methylation level, we then measured the whole m6A contents. Compared with that of the 22PDL group, the m6A methylation level of the 35PDL group and the 49PDL group reduced, with decreases of 35.1% and 52.3%, respectively (*p* < 0.05). The m6A methylation level of the PSp group decreased by 41.1% (*p* < 0.05), but no significant change occurred in the PSi group (*p* > 0.05) compared with that of the 22PDL group. Moreover, the difference of m6A RNA methylation level in the PSp group was not statistically significant compared with that in the 49PDL group (*p* > 0.05), as shown in Figure 2A,B.

Subsequently, we measured the whole activity of RNA methyltransferases and demethylases separately. Compared with that in the 22PDL group, RNA methyltransferases activity decreased significantly in the 35PDL, 49PDL, PSi, and PSp groups (*p* < 0.05). Compared with that in the 49PDL group, RNA methyltransferases activity increased in the PSi group (*p* < 0.05), while it was not statistically significant in the PSp and 49PDL group (*p* > 0.05). Therefore, the activity of RNA methyltransferases decreased during cellular senescence. In the PSi group, at the beginning of premature senescence, RNA methyltransferases activity decreased less from the results of Figure 2C. According to Figure 2D, the RNA demethylases activity showed no significant difference among all groups (*p* > 0.05). Therefore, in the premature senescence initiation stage of the PSi group, the accumulated ROS did not change the m6A level and enzymes activity in a short period and they were still similar to those of the 22PDL group. With the development of premature senescence, the RNA methyltransferases activity and m6A level gradually decreased in the PSp group with an important role. In the process of senescence, the RNA methyltransferases played a leading role in cellular replicative or premature senescence with decreased levels at senescent stage.

### 2.3. Oxidative Stress Affects the m6A-Methylation-Modulated System during Premature Senescence

Further, the expression levels of RNA methyltransferases, demethylases, and RNA methylation binding proteins were detected, respectively, during cellular senescence. The results for RNA methyltransferases METTL3, METTL14, METTL16, WTAP, and KIAA1429; the demethylases FTO and ALKBH5; and methylation binding proteins YTHDC1, YTHDC2, YTHDF1, YTHDF2, YTHDF3, HNRNPA2B1, and HNRNPC are presented in Figure 3 separately.

For mRNA expression, compared with that in the 22PDL group, *METTL3* increased, while *METTL16* and *WTAP* decreased in the 35PDL, 49PDL, PSi, and PSp groups significantly (*p* < 0.05), and *KIAA1429* increased in the PSi and PSp groups (*p* < 0.05). Compared with that in the 49PDL group, the level of *METTL3*, *WTAP*, and *KIAA1429* increased in the PSp group, respectively (*p* < 0.05). *METTL14* revealed no significant difference in each group *(p* > 0.05). About RNA demethylases, the levels of *FTO* and *ALKBH5* decreased in senescent cells, and they were lower in the PSp group than in the 49PDL group (*p* < 0.05). In addition, the mRNA expressions of the six binding proteins, *YTHDC1/2*, *YTHDF1/3*, *HNRNPA2B1*, and *HNRNPC* showed no significant differences in both the 49PDL and PSp groups (*p* > 0.05), while only *YTHDF2* decreased (*p* < 0.05) from Figure 3A.

For protein expression, compared with that in the 22PDL group, METTL14 and METTL16 increased significantly in the 35PDL, 49PDL, PSi, and PSp groups, but KIAA1429 decreased (*p* < 0.05), as shown in Figure 3B. METTL14, METTL3, WTAP, and KIAA1429 were higher in the PSp group than that in the 49PDL, respectively, while METTL16 was lower than in the 49PDL group (*p* < 0.05). As for RNA demethylases, FTO and ALKBH5 decreased in senescent cells, and ALKBH5 was lower in the PSp group than that in the 49PDL group, while FTO was lower in the 49PDL group than that in the PSp group (*p* < 0.05). About the RNA methylation binding proteins, YTHDF3 increased with senescence, while each of the other six proteins decreased significantly (*p* < 0.05). The expression of YTHDC1 and YTHDF1/2/3 was higher in the 49PDL group than that of the PSp group, respectively (*p* < 0.05). After that, we also detected the distribution and the relative level of YTHDF3 by immunofluorescence, and found that it was distributed in the nucleus and cytoplasm and might bind to the m6A site, and be involved in RNA translation and decay [21]. The YTHDF3 protein level was increased in senescent cells as shown in Figure 3C.

Thus, the expressions of METTL14, METTL16, and YTHDF3 increased in senescent cells, while they decreased in KIAA1429, FTO, ALKBH5, YTHDC1, YTHDC2, YTHDF1, YTHDF2, HNRNPA2B1, and HNRNPC, respectively. In replicative and premature senescent cells, the expression of METTL14, METTL3, WTAP, and KIAA1429 and FTO in the PSp group was higher than that in the 49PDL group, while the expression of METTL16, ALKBH5, YTHDC1, and YTHDF1/2/3 in the PSp group was lower than that in the 49PDL group (*p* < 0.05). The PSi and PSp group represented the specific stages of premature senescence with differential levels of m6A-methylation-related enzymes and binding proteins. Taken together, these changes formed a specific expression profile to cellular senescence regulated by RNA methylation.

### 2.4. Cellular ROS Level Is Connected with m6A RNA Methylation Profile

In order to verify the effects of ROS on m6A profile, we conducted the correlation analyses of ROS correlated with m6A content or RNA methyltransferases activity. They included replicative senescence and premature senescence, using the 22PDL, 35PDL, and 49PDL groups for replicative senescence series, and the 22PDL, PSi, and PSp groups for premature senescence series, respectively. Correlation analyses were conducted to distinguish the differences between the two series. We found that ROS levels were negatively correlated (*r* = −0.947, *p* = 0.004) with m6A content in the replicative senescence series in Figure 4A, but there was no correlation in the premature senescence series (Appendix A). ROS levels were negatively correlated (*r* = −0.827, *p* = 0.042) with RNA methyltransferases activity in the replicative senescence series, as shown in Figure 4A, but not associated with the premature senescence series (Appendix A).

Subsequently, to study the effects of ROS on m6A regulation system, we analyzed the internal correlations between ROS level and each RNA methyltransferase, demethylase, and methylated binding protein accordingly. ROS levels were negatively correlated with the protein levels of METTL3 (*r* = −0.873, *p* = 0.002), KIAA1429 (*r* = −0.991, *p* < 0.001), FTO (*r* = −0.922, *p* < 0.001), ALKBH5 (*r* = −0.930, *p* < 0.001), YTHDC1 (*r* = −0.797, *p* = 0.013), YTHDC2 (*r* = −0.0895, *p* = 0.001), YTHDF1 (*r* = −0.955, *p* < 0.001), YTHDF2 (*r* = −0.949, *p* < 0.001), and HNRNPC (*r* = −0.773, *p* = 0.015) in the replicative senescence series separately, while ROS levels were positively correlated with the protein levels of METTL16 (*r* = 0.989, *p* < 0.001) and YTHDF3 (*r* = 0.946, *p* < 0.001). In the premature senescence series, ROS levels were negatively correlated with protein levels of METTL3 (*r* = −0.797, *p* = 0.013) and YTHDC1 (*r* = −0.873, *p* = 0.002), but positively correlated with protein levels of METTL16 (*r* = 0.705, *p* = 0.034), YTHDC2 (*r* = 0.898, *p* = 0.001), YTHDF3 (*r* = 0.982, *p* < 0.001), and HNRNPA2B1 (*r* = 0.878, *p* = 0.002), and no correlation was found with protein levels of METTL14 and WTAP (Figure 4B,C; Appendix A).

Therefore, except for YTHDC2, the correlations between ROS level and the protein level of methylation-related enzymes were similar in both series of senescent cells. ROS could affect the levels of m6A RNA methylation-related enzymes directly, which might also modulate the m6A-modified genes, contributing to cell senescence.

### 2.5. The m6A Modification Regulates Its Targeted Genes Related to Senescence

Then, we first focused on FOXO, the important determinant in senescence, to select the upstream, downstream, and cross-connection genes related to senescence or oxidative stress for observing their expression patterns and intrinsic network relationships. They included SIRT3, FOXO1, MST1, IRS2, E2F3, ADCY9, PRKACB, CREB1, and PER2, respectively. Next, we observed the m6A regulation status of the targeted genes. The expression profiles of the nine selected genes above are shown in Figure 5.

The interactions of enzymes associated with m6A methylation and genes related to senescence were analyzed by string database. The PPI network is shown in Figure 6A. The candidate genes related to cell senescence were associated with m6A methylation enzymes and binding proteins, which formed a complex web to cellular senescence. SIRT3 and FOXO1, as key proteins, were associated with more networks and interacted with other related proteins.

For mRNA expression, compared with that in the 22PDL group, the level of *MST1* increased in either senescent group, while the other seven genes decreased significantly (*p* < 0.05). *PRKACB* had no significant difference (*p* > 0.05). Compared with that in the 49PDL group, *IRS2* increased in the PSp group, and *SIRT3*, *E2F3*, *ADCY9*, *PRKACB*, *CREB1*, and *PER2* decreased separately in the PSp group (*p* < 0.05), as shown in Figure 5A.

For the protein expression, compared with that in the 22PDL group, SIRT3 and E2F3 increased in senescent cells; MST1, ADCY9, PRKACB, CREB1, and PER2 decreased in the 49PDL and PSp groups; and IRS2 decreased in the 49PDL group and increased in the PSp group (*p* < 0.05). Compared with that in the 49PDL group, the level of IRS2 increased in PSp, while SIRT3, E2F3, ADCY9, PRKACB, CREB1, and PER2 decreased in PSp (*p* < 0.05). FOXO1 showed no significant difference in each group (*p* > 0.05) (Figure 5B).

According to the above results, the mRNA and protein expressions of SIRT3, E2F3, FOXO1, MST1, and IRS2 were inconsistent, which might be regulated by specific m6A methylation styles in the senescence process. Therefore, MeRIP-qPCR was further used to detect their m6A methylation modification abundance. Shown in Figure 6B, the m6A methylation modification abundance of SIRT3 and E2F3 was higher in the PSp group than that in the 22PDL and 49PDL group (*p* < 0.05). However, the m6A modification abundance of IRS2 in the 49PDL group was higher than that in the PSp group (*p* < 0.05). The other two genes did not differ statistically between the two kinds of senescent cells (*p* > 0.05). Thus, RNA m6A modification inhibited the posttranscriptional translation of SIRT3, E2F3, and IRS2, causing reduced protein levels. Additionally, m6A methylation modifications did not affect the protein expression of FOXO1 and MST1 between the 49PDL and PSp group. In the genes web related with cellular senescence, m6A methylation also modified the specific targeted genes under the specific epitranscriptome microenvironment.

## 3. Discussion

Based on our replicative and premature senescence models, the first comprehensive map of RNA methylation microenvironment status has been obtained related to cell senescence, including changes in the m6A level, enzymes activity, and expression profile of the RNA methylation regulatory system. ROS in senescent cells affects the expression of m6A-modification-related enzymes and the binding proteins, also regulating the targeted genes associated with senescence, and composed of important molecular events.

Accumulation of cell senescence may lead to irreversible deterioration of organ dysfunction and metabolic decline gradually. When HEFs were in the young and mid-aged group, the cells had high proliferation ability in a vigorous style. However, when the cells were continuously cultured to around the 49PDL, they grew slowly or even stopped proliferating and showed signs of senescence. Aside from cell cycle arrest, senescent cells underwent morphological, biochemical, and functional changes, which were signs of cellular senescence [22]. H_2_O_2_ is usually used as the subject to induce the oxidative stress model. In this study, 400 μmol/L H_2_O_2_ was used to treat young cells, inducing premature senescence of HEFs. In the process of replicative senescence, the total intracellular ROS level elevated with the increase in generation age. After H_2_O_2_ treatment, the young cells obtained high and sharp ROS level, and then gradually decreased irreversibly, resulting in persistent premature senescence. We have confirmed that the cellular replicative senescence comes from the results of the accumulation of intracellular ROS and the imbalance of ROS metabolism. When young cells were stimulated by external factors, the amount of ROS increased in a short time [23]. With the aggravation of environmental pollution, cells are exposed to both exogenous and endogenous oxides, with increased ROS production and damages.

ROS signaling plays an essential role in epigenetic processes such as DNA methylation and histone modification from the studies in vitro and in vivo [24,25], inducing general genome hypomethylation and specific DNA promoter hypermethylation by upregulating DNA methyltransferase and its complexes. The biological functions of m6A are mainly regulated by RNA methyltransferases, demethylases, and binding proteins jointly [16]. Young cells had less endogenous ROS level before H_2_O_2_ treatment. After that, the exogenous ROS increased sharply, and the global m6A level decreased, which was caused by the decrease of RNA methyltransferases activity. The cells’ epigenetic microenvironment showed different changes in the protein expression levels of METTL14, WTAP, FTO, ALKBH5, HNRNPC, YTHDC1/2, and YTHDF1/2, also with HNRNPA2B1 decreasing and the METTL14, METTL16, METTL3, YTHDF3, and KIAA1429 increasing, which regulated the m6A methylation level and direction during cellular senescence. To the PSi and PSp group, there are some different indicators, including ROS level, SA-β-gal, m6A level, and the related enzymes activity, respectively, which might be associated with the differential oxidative stress levels and interactions with epigenetic effects. These results have suggested that ROS and RNA methylation play particularly important roles during cellular premature senescence. Another recent study has shown that RNA m6A modification can correspond to the damage response caused by ROS [26], which is consistent with our results.

Additionally, the m6A modification level of senescence related with genes *SIRT3*, *IRS2*, and *E2F3* was different in the two kinds of senescent cells, respectively, and the protein level also increased or decreased at different m6A levels. These different changes of RNA m6A modification in senescent cells may promote methylation or demethylation of mRNA cooperatively, mediating corresponding changes in the body. METTL16 is the RNA methyltransferase, existing as a monomer, and has different recognition mechanisms for different RNA substrates related to the shearing of precursor mRNA [27,28,29]. In this study, the protein expression of METTL3 in the PSp group increased more obviously than that in the 49PDL group. However, there was no significant difference of METTL14 in senescent cells, which might be because METTL14 mainly supported METTL3 structurally and played a key role in the recognition of RNA substrates [30,31,32]. Furthermore, WTAP interacts with METTL3 and METTL14, and requires for their localization into nuclear speckles enriched with pre-mRNA processing factors and for catalytic activity of the m6A methyltransferase in vivo [33]. Therefore, the combined actions of WTAP–METTL3/METTL14 weakened the methyltransferase activity in the PSp group. Furthermore, m6A methyltransferases activity was the highest in young cells, basically consistent with the protein level of RNA methyltransferase. There were no significant differences in RNA demethylases activity between young and senescent cells. According to the correlation analysis, ROS inhibited RNA methyltransferases activity. Besides, in the premature senescence series with exogenous H_2_O_2_, ROS only temporarily increased, which might affect other signaling pathways. It can be concluded that both acute and chronic ROS exposure could directly promote METTL16 and YTHDF3 protein expression. So, ROS can directly affect the enzyme activity and protein expression level of RNA methyltransferase, especially METTL16, leading to m6A modification of senescence-related genes; then, YTHDF3 binds to the m6A sites to regulate the translation of targeted genes.

The m6A modification is widespread in the whole transcriptome. The increase in m6A methylation sites in cells during senescence may lead to a posttranscriptional functional block of related genes and, thus, promote senescence. Methylated mRNAs often contain multiple m6A methylation sites [34]. The m6A sites are enriched in the vicinity of the stop codon and in the 3′UTR, and conserved in the human and mouse transcriptome. The 3′UTR is an important region for RNA regulation, affecting RNA stability, subcellular localization, and translation regulation. Some of these events are regulated by RNA-binding proteins, which bind to cis-acting structural mods or common sequences within the 3′UTR to coordinate the effects of RNA processing [35].

Currently, studies have consistently revealed FOXO (Forkhead box O) transcription factors’ function as important determinants in aging and longevity [36]. Therefore, we took FOXO as the node, selected the signaling pathways related to senescence, and detected the genes levels of its upstream and downstream, including the longevity regulating pathway (hsa04211), FOXO signaling pathway (hsa04068), cellular senescence pathway (hsa04218), and circadian entrainment pathway (hsa04713). They have shown the complex network and interacted with each other, in relation to m6A methylation or oxidative stress. Their roles in longevity determination are attractive and need to be fully elucidated. In this study, the protein levels of SIRT3 in replicative and premature senescent cells increased and the mRNA levels had the opposite changes. Meanwhile, the protein expression in the premature senescence group was lower than that in the replicative senescence group, while the m6A methylation level was opposite. Therefore, RNA hypermethylation of SIRT3 can promote its posttranscriptional translation, thus causing premature senescence. Although the SIRT3 protein level increased after translation in premature senescence, it was different from that of replicative senescent cells in general, which might be an internal factor for different senescence types. SIRT3 is a family of NAD^+^ dependent histone deacetylases that is homologous to SIRT2, involved in calorie-restriction-related longevity extension in organisms [37]. The attenuation or ablation of SIRT3 is related to the accelerated development of various aging diseases [38]. The methylation level of *E2F3* was higher in premature senescence cells than that in the young cells, and the protein expression was consistent with that of the young group. Currently, the regulatory role of *E2F3* in the senescence process is unclear, but its m6A modification is involved in miRNA/*E2F3*-mediated proliferation inhibition of gastric cancer cells [38], thus suggesting that m6A modification in *E2F3* may also be an important factor of cell senescence. The methylation level of *E2F3* was higher in the PSp group than that in the 49PDL group, and the increased methylation level promoted the protein expression, which might function as the cause of cellular premature senescence. FOXO1 and IRS2 play important roles in the occurrence and treatment of tumors as well as in neurodegenerative diseases [36,39], but the association between their m6A modification and aging has rarely been reported. In our study, there were no significant differences between the methylation level and protein expression level of FOXO1 in all groups, and no change in FOXO1 was observed during replicative and premature senescence. These changes might be because other members of the FOXO family played roles in senescence. FOXO3 could inhibit the expression of oncogene FOXM1, restrict stem cell renewal, and trigger senescence [40]. The methylation level of IRS2 was higher in the 49PDL group than in the 22PDL and PSp groups, and the protein expression was lower in the 49PDL group than in the PSp group. Hence, the m6A hypermethylation of IRS2 inhibits the protein expression in replicative senescent cells. After H_2_O_2_ treatment, the degree of methylation of *IRS2* is reduced and the protein expression increased, thereby taking part in the occurrence of premature senescence and acting as the internal reasons of different molecules in replicative and premature senescence.

Collectively, the cellular endogenous and exogenous ROS induce cellular senescence via m6A modification, with a specific and differential profile of m6A-related enzymes, binding proteins, and the targeted genes. ROS contents also modulate the overall m6A levels and change the epigenetic microenvironment, contributing to the occurrence of cell senescence. From the present researches, the RNA m6A methylation modification open a new window to interpreting the mechanisms for the occurrence and development of senescence and the related diseases. The reversible levels of m6A and the modified proteins will bring about the potential possibility for the prevention and intervention of cell senescence as the targets in the future. The interactions of exogenous oxidative stress, RNA methylation microenvironment, and the key genes related with senescence have enriched the epitranscriptomics theory of cell senescence to a better extent.

## 4. Materials and Methods

### 4.1. Cell Culture and H_2_O_2_ Treatment

HEFs were obtained from Chinese Academy of Medical Sciences, Institute of Basic Medicine and cultured in Low-glucose Dulbecco’s modified Eagle’s medium (L-DMEM, Gibco, Grand Island, NE, USA) with 10% fetal bovine serum, 100 U/mL penicillin, and 0.1 mg/mL streptomycin (Gibco, Grand Island, NE, USA). Constant temperature incubator (Thermo Fisher, Waltham, MA, USA) conditions were strictly controlled at 37 °C, 95% relative humidity, and 5% CO_2_. HEFs were continuously passaged and stopped proliferating around 52 population doubling level (PDL). According to the age definition in cell culture, we divided them into young cells (22PDL), mid-aged (35PDL), and replicative senescence cells (49PDL).HEFs around 22PDL were exposed to 400 μmol/L H_2_O_2_ (BDH Chemicals Ltd., Poole, England) for 2 h every day for 4 consecutive days; the cells group was called the premature senescence initiation group (PSi) and then given fresh L-DMEM with fetal bovine serum and cultured for another 7 days, defined as the premature senescence persistence group (PSp), as previously described [20].

### 4.2. SA-β-Galactosidase Staining

The SA-β-galactosidase staining kit (Beyotime, Shanghai, China) was used to detect cell senescence. Cells were washed with PBS and fixed with fixative solution for 15 min. Fixed cells were washed twice with PBS and stained with staining solution, containing staining supplement and X-Gal, then left overnight at 37 °C in an incubator. The 6-well plates were covered with plastic wrap to prevent evaporation. The SA-β-gal positive blue cells were visualized and photographed under a microscope (Leica, Solms, Germany), and the number of staining cells was recorded. The positive staining rate indicated the proportion of blue cells in the group.

### 4.3. Measurement of Total ROS in Cells

The total cellular ROS level was measured by Reactive oxygen species (ROS) detection assay kit (Biovision, Milpitas, CA, USA). After entering the cells, H_2_DCFDA was modified by cellular esterase to form nonfluorescent H_2_DCF, and then H_2_DCF was oxidized by the intracellular ROS to produce high-fluorescence products. In brief, 3 × 10^5^ cells were seeded into the 12-well plates, overnight. Meanwhile, H_2_DCFDA (ROS Label) diluted with serum-free medium was added to each well and incubated in dark at 37 °C for 45 min. Multifunction microplate reader (BIO⁃TEK, Winooski, VT, USA) was used to measure the mean of fluorescence intensity (MFI) at Ex/Em = 495/529 nm. Then, the total protein of each group was extracted and the concentration was measured using the BCA method. The total cell ROS level was quantitated relatively using the formula as follows:(1)$$\mathrm{ROS}\;\mathrm{level}=\frac{\mathrm{MFI}}{\mathrm{average}\;\mathrm{protein}\;\mathrm{mass}}-\frac{\mathrm{MFI}\;\mathrm{of}\;\mathrm{the}\;\mathrm{control}\;\mathrm{well}}{\mathrm{average}\;\mathrm{protein}\;\mathrm{mass}\;\mathrm{of}\;\mathrm{the}\;\mathrm{control}\;\mathrm{well}}$$

### 4.4. Real-Time RT-PCR for mRNA Level

The total RNA was extracted by Trizol Reagent (Invitrogen, Carlsbad, CA, USA). The cDNA was synthesized from total RNA using PrimeScipt Master Mix (TaKaRa, Kyodo, Japan). The mRNA expression level was quantified by qPCR using a SYBR Premix Ex Taq™ II Kit (Tli RNaseH Plus, TaKaRa, Kyodo, Japan) and the fluorescence quantitative PCR instrument (CFX connect, BIO-RAD, Hercules, CA, USA). The primers used are shown in Table 1.

### 4.5. Western Blot for Protein Level

Cells were harvested after being washed in PBS. The total protein was extracted via RIPA Lysis Buffer (Beyotime, Shanghai, China) with a protease inhibitor PMSF (Beyotime, Shanghai, China). Proteins from whole lysates were separated on 10% SDS-polyacrylamide gels electrophoresis (SDS-PAGE, Beyotime, Shanghai, China) and then transferred to polyvinylidene difluoride membranes (Millipore Corporation, Burlington, MA, USA) separately. Following incubation with 5% nonfat milk for 1 h, the bolts were incubated with the primary antibody of METTL3 (ab195352, Abcam, Cambridge, UK), METTL14 (ab98166, Abcam, Cambridge, UK), METTL16 (ab186012, Abcam, Cambridge, UK), WTAP (ab195380, Abcam, Cambridge, UK), KIAA1429 (25712-1-AP, Proteintech, Rosemont, IL, USA), FTO (ab124892, Abcam, Cambridge, UK), ALKBH5 (ab195377, Abcam, Cambridge, UK), YTHDC1 (ab220159, Abcam, Cambridge, UK), YTHDC2 (ab220160, Abcam, Cambridge, UK), YTHDF1 (17479-1-AP, Proteintech, Rosemont, IL, USA), YTHDF2 (24744-1-AP, Proteintech, Rosemont, IL, USA), YTHDF3 (25537-1-AP, Proteintech, Rosemont, IL, USA), HNRNPA2B1 (14813-1-AP, Proteintech, Rosemont, IL, USA), HNRNPC (ab133607, Abcam, Cambridge, UK), SIRT3 (ab217319, Abcam, Cambridge, UK), E2F3 (ab50917, Abcam, Cambridge, UK), FOXO1 (ab52857, Abcam, Cambridge, UK), IRS2 (ab134101, Abcam, Cambridge, UK), MST1 (14946, CST, Danvers, MA, USA), ADCY9 (ab191423, Abcam, Cambridge, UK), PRKACB (12232-1-AP, Proteintech, Rosemont, IL, USA), CREB1 (ab32515, Abcam, Cambridge, UK), PER2 (ab179813, Abcam, Cambridge, UK), and ß-actin (ab8226, Abcam, Cambridge, UK), respectively, overnight at 4 °C. The dilution rate of the primary antibody was 1:1000. The membrane was triply washed with TBST at room temperature for 10 min and then treated with the corresponding second antibody, goat antirabbit antibody (ab6721, Abcam, Cambridge, UK) or goat antimouse antibody (ab6789, Abcam, Cambridge, UK) at 1:5000 dilution for 1 h. Proteins in the membrane were detected by automatic chemiluminescence imaging analysis system (Tanon-5200, Beijing, China) with Chemiluminescent HRP Substrate (Millipore, MA, USA).

### 4.6. Global RNA m6A Content Quantification

Total RNA was extracted via TRIzol (Invitrogen, Carlsbad, CA, USA) and RNA quality was assessed by NanoDrop (Thermo Fisher Scientific, Farmingdale, NY, USA). The m6A modification level of total RNA was examined according to the instruction of EpiQuikTM m6A RNA methylation quantification kit (Colorimetric, EpiGentek, Farmingdale, NY, USA). In brief, 200 ng RNA was added to the assay well; then, the appropriate diluted concentration of detection antibody solution was added to the assay well. The m6A levels were quantified using the colorimetrical analysis of absorbance at 450 nm and calculated according to the standard curve.

### 4.7. The Activity of m6A-Modification-Related Enzymes

The total activity of m6A-modification-related enzymes was determined by colorimetry including RNA methyltransferase and demethylases (EpiGentek, Farmingdale, NY, USA). In the assay, the unique substrate was stably coated on the strip wells. Active m6A methyltransferase/demethylase binding to methylated adenosine residues was contained in the substrate. The methylated m6A in the substrate could be recognized by a high-affinity m6A antibody and the immunosignal was enhanced with enhancer solution. The ratio or amount of methylated m6A, which was proportional to enzyme activity, could then be colorimetrically quantified through an ELISA-like reaction. At the wavelength of 450 nm, the absorbance was read within 2~10 min with a microplate analyzer.
(2)$$\begin{array}{l} \mathrm{The}\;\mathrm{activity}\;\mathrm{of}\;\mathrm{methylases}\;\left(\mathrm{ng}/h/\mathrm{mg}\right)\\ =\frac{\left(\mathrm{OD}\;\mathrm{sample}-\mathrm{OD}\;\mathrm{blank}\right)\times 1000}{\mathrm{The}\;\mathrm{slope}\;\mathrm{of}\;\mathrm{the}\;\mathrm{standard}\;\mathrm{curve}\;\times \;\mathrm{Nucleoprotein}\;\mathrm{mass}\;\left(\mu g\right)\times \;\mathrm{Enzymatic}\;\mathrm{reaction}\;\mathrm{time}\;\left(h\right)} \end{array}$$
(3)$$\begin{array}{l} \mathrm{The}\;\mathrm{activity}\;\mathrm{of}\;\mathrm{demethylases}\;\left(\mathrm{ng}/h/\mathrm{mg}\right)\\ =\frac{\left(\left(\mathrm{OD}\;\mathrm{control}-\mathrm{OD}\;\mathrm{blank}\right)-\left(\mathrm{ODsample}-\mathrm{OD}\;\mathrm{blank}\right)\right)\times 1000}{\mathrm{The}\;\mathrm{slope}\;\mathrm{of}\;\mathrm{the}\;\mathrm{standard}\;\mathrm{curve}\;\times \;\mathrm{Nucleoprotein}\;\mathrm{mass}\;\left(\mu g\right)\times \;\mathrm{Enzymatic}\;\mathrm{reaction}\;\mathrm{time}\;\left(h\right)} \end{array}$$

### 4.8. MeRIP-qPCR for m6A Modification Abundance of Aging-Related Gene

The Magna MeRIP m6A assay kit (Milipore, MMAS, Burlington, MA, USA) was used to obtain m6A-captured RNA. In the MeRIP assay, RNA was chemically fragmented into 100 bp nucleotides or smaller fragments followed by magnetic immunoprecipitation with a monoclonal antibody toward m6A. After immunoprecipitation, the RNA fragments were eluted and purified with the RNeasy mini kit (Qiagen, Hilden, Germany). Finally, the isolated RNA fragments could be subjected to qPCR. Primers were designed based on sequencing results and methylation sites retrieved from MeT-DB database (The MethylTranscriptome Database, http://compgenomics.utsa.edu/methylation/, accessed on 4 January 2021.) [41]. *SIRT3*, *FOXO1*, *IRS2*, and *E2F3* primers were designed in the 3′UTR regions, and primers for *MST1* were designed in the CDS region. The screened 5 genes associated with senescence are verified for their m6A methylation modification and the primers used are shown in Table 2.

### 4.9. Protein–Protein Interaction (PPI) Network

The genes were imported into the STRING database, containing comprehensive information about the interactions among proteins. The PPI network was constructed based on importing the data into Cytoscape 3.9.1 software (National Institute of General Medical Sciences (NIGMS), Seattle, WA, USA); then, the network was analyzed by Network Analyzer.

### 4.10. Statistical Analysis

All experiments were performed at least three times. All values and error bars in graphs were means ± SEM. Analysis of variance (ANOVA) was used to analyze the differences between experimental groups and control (GraphPad Software Inc., San Diego, CA, USA).

Pearson correlation was used to analyze the correlation between ROS level and protein expression level, enzyme activity, or m6A content (SPSS 19.0). The differences between the two groups were tested by Bonferroni. *p* < 0.05 was considered statistically significant.

## Figures and Tables

**Figure 1 ijms-23-09271-f001:**
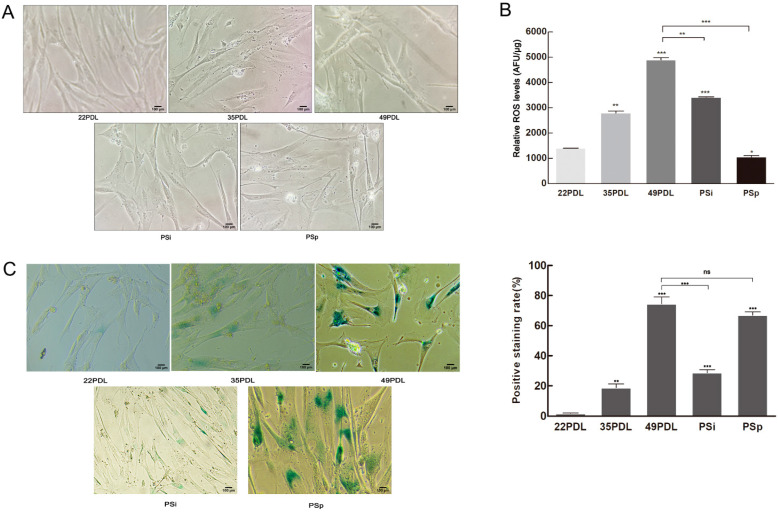
SA-β-gal staining and ROS level. (**A**) Cell morphology under the microscope (200×, 100 μm). (**B**) ROS level in the cells. (**C**) The SA-β-gal staining and the blue staining rate. The β-galactosidase was dyed blue in senescent cells (200×, 100 μm). The data are presented as the mean ± SEM (*n* = 3). Compared with the 22PDL or 49PDL group, * *p* < 0.05, ** *p* < 0.01, *** *p* < 0.001, or ns *p* > 0.05.

**Figure 2 ijms-23-09271-f002:**
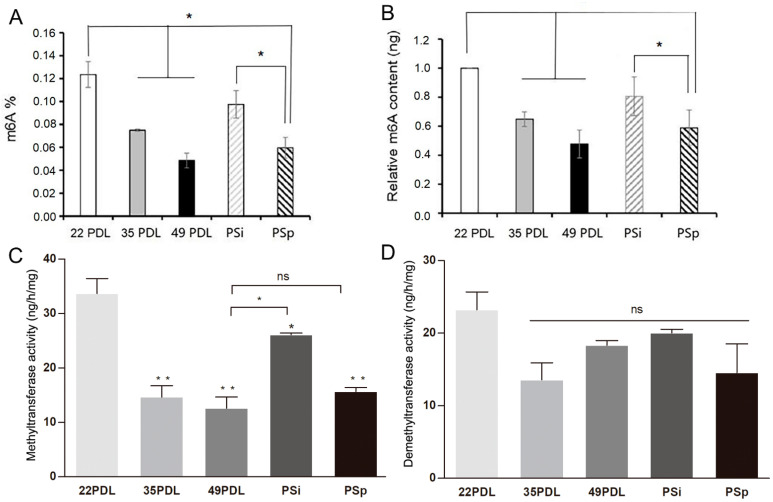
The overall m6A RNA methylation level and activities of enzymes associated with m6A modification. (**A**) The proportion of m6A-modified mRNA in 200 ng total RNA. (**B**) The relative m6A contents in each group. (**C**) The activity of methyltransferases, ng/h/mg. (**D**) The activity of demethyltransferases, ng/h/mg. The data are presented as the mean ± SEM (*n* = 3). Compared with the 22PDL or 49PDL group, * *p* < 0.05, ** *p* < 0.01, or ns *p* > 0.05.

**Figure 3 ijms-23-09271-f003:**
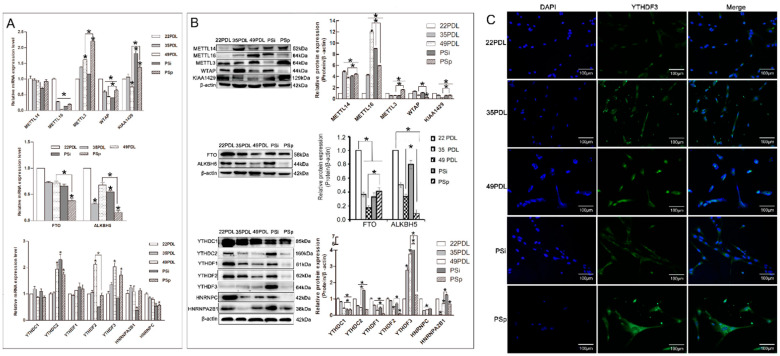
Expression of enzymes associated with m6A modification. (**A**) The relative mRNA expression of RNA methyltransferase, demethylase, and binding proteins. The 22PDL group was set as 1 and the values of the other groups were relative to the 22PDL group. (**B**) Western blotting analysis for RNA methyltransferases, demethylases, and binding proteins and their quantifications. (**C**) The immunofluorescence images of YTHDF3 under the fluorescence microscope (200×, 100 μm). The green represented immunofluorescence stained by the YTHDF3. Nuclei were counterstained with DAPI as blue. The data are presented as the mean ± SEM (*n* = 3). Compared with the 22PDL or 49PDL group, * *p* < 0.05.

**Figure 4 ijms-23-09271-f004:**
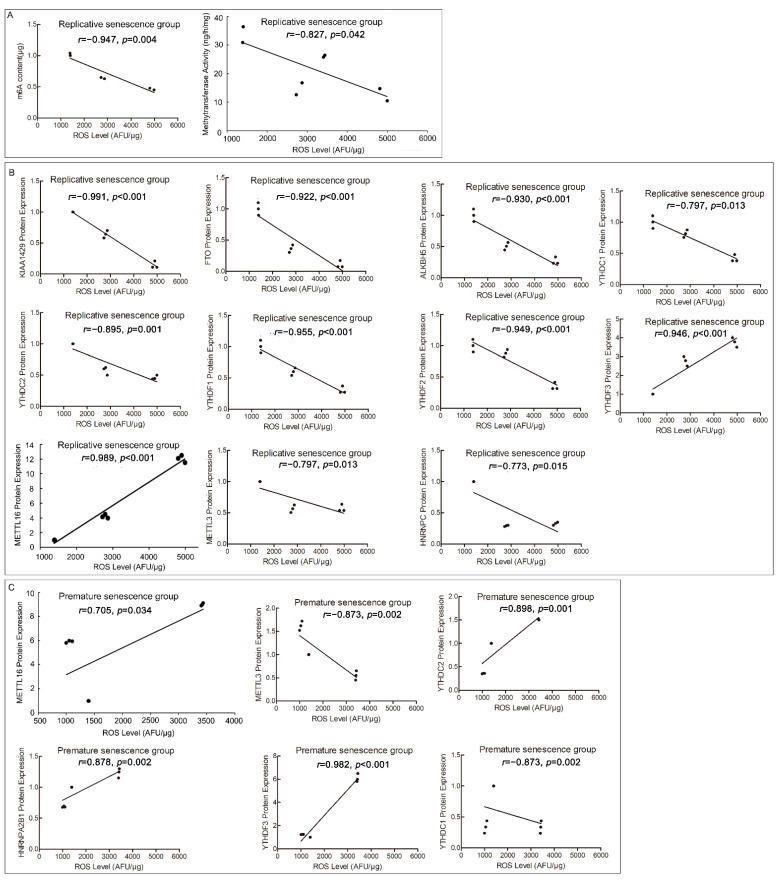
Correlation analysis of ROS and m6A RNA methylation indexes. (**A**) Correlation analysis of ROS with m6A contents and RNA methyltransferase activity from the 22PDL, 35PDL, and 49PDL group. (**B**) Correlation analysis of ROS with RNA methyltransferases, demethylases, and binding proteins in replicative senescence group from the 22PDL, 35PDL, and 49PDL groups. (**C**) Correlation analysis of ROS with RNA methyltransferases, demethylases, and binding proteins in premature senescence group from the 22PDL, PSi, and PSp groups, respectively.

**Figure 5 ijms-23-09271-f005:**
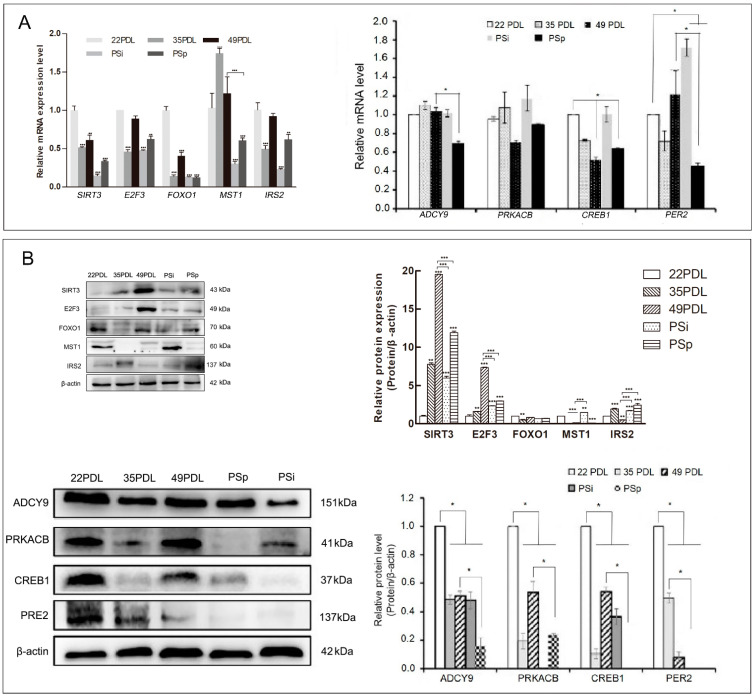
Expression of candidate genes related to senescence. (**A**) The mRNA expression of senescence-related genes. (**B**) Western blotting analysis for candidate genes related to senescence and protein level quantification. The data are presented as the mean ± SEM (*n* = 3). Compared with the 22PDL or 49PDL group, * *p* < 0.05, ** *p* < 0.01, or *** *p* < 0.001.

**Figure 6 ijms-23-09271-f006:**
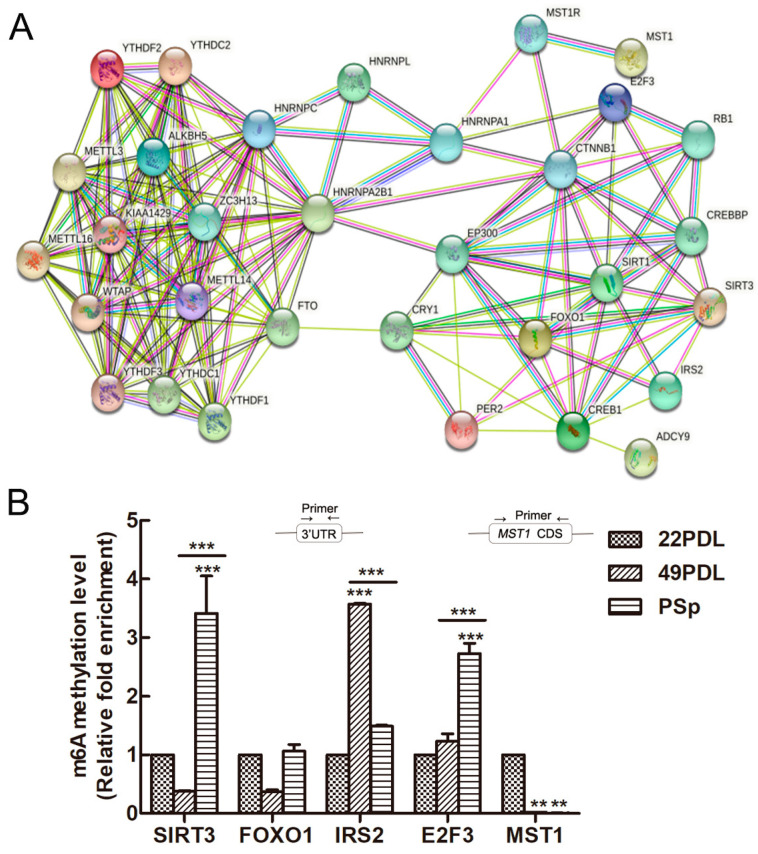
The PPI network and the m6A methylation of candidate genes related to senescence. (**A**) The PPI network included the regulation of m6A RNA-related enzymes and senescence-related proteins. (**B**) Methylation levels in specific regions of aging-related genes detected by MeRIP-qPCR. The 22PDL group was set as 1 and the methylation level of the other group was relative to the 22PDL group. The data are presented as the mean ± SEM (*n* = 3). Compared with the 22PDL or 49PDL group, ** *p* < 0.01, or *** *p* < 0.001.

**Table 1 ijms-23-09271-t001:** The primer sequences used in this study for qPCR.

Primer Name	Forward (5′-3′)	Reverse (5′-3′)
**Methyltransferases**		
*METTL16*	TGGAGCAACCTTGAATGGCTGG	CCATCAGGAGTGTCTTCTGTGG
*METTL14*	CTGAAAGTGCCGACAGCATTGG	CTCTCCTTCATCCAGATACTTACG
*METTL3*	CTATCTCCTGGCACTCGCAAGA	GCTTGAACCGTGCAACCACATC
*WTAP*	GCAACAACAGCAGGAGTCTGCA	CTGCTGGACTTGCTTGAGGTAC
*KIAA1429*	TGACCTTGCCTCACCAACTGCA	AGCAACCTGGTGGTTTGGCTAG
**Demethylases**		
*FTO*	CCAGAACCTGAGGAGAGAATGG	CGATGTCTGTGAGGTCAAACGG
*ALKBH5*	CCAGCTATGCTTCAGATCGCCT	GGTTCTCTTCCTTGTCCATCTCC
**Binding proteins**		
*YTHDF1*	CAAGCACACAACCTCCATCTTCG	GTAAGAAACTGGTTCGCCCTCAT
*YTHDF2*	TAGCCAGCTACAAGCACACCAC	CAACCGTTGCTGCAGTCTGTGT
*YTHDF3*	GCTACTTTCAAGCATACCACCTC	ACAGGACATCTTCATACGGTTATTG
*HNRNPA2B1*	CAGCAACCTTCTAACTACGGTCC	CACTGCCTCCTGGACCATAGTT
*HNRNPC*	TGGGCTGCTCTGTTCATAAGGG	CTCGGTTCACTTTTGGCTCTGC
*YTHDC1*	TCAGGAGTTCGCCGAGATGTGT	AGGATGGTGTGGAGGTTGTTCC
*YTHDC2*	GAAAGCTCCTGAACCTCCACCA	GGTTCTACTGGCAAGTCAGCCA
**Aging-related genes**		
*E2F3*	AGCGGTCATCAGTACCTCTCAG	TGGTGAGCAGACCAAGAGACGT
*FOXO1*	CTACGAGTGGATGGTCAAGAGC	TCTACGAGTGGATGGTGCGTTG
*IRS2*	CCTGCCCCCTGCCAACACCT	TGTGACATCCTGGTGATAAAGCC
*MST1*	CTGTGTAGCAGACATCTGGTCC	CTGGTTTTCGGAATGTGGGAGG
*SIRT3*	CCCTGGAAACTACAAGCCCAAC	GCAGAGGCAAAGGTTCCATGAG
*ADCY9*	CTCAAAACGGCTGCCAAGACGA	GAGAAGTCTGACTGTTGGTGAGC
*PRKACB*	GCAGTGGATTGGTGGGCATTAG	ACTGAAGTGGGATGGGAATCGG
*CREB1*	GACCACTGATGGACAGCAGATC	GAGGATGCCATAACAACTCCAGG
*PER2*	AGCTGCTTGGACAGCGTCATCA	CCTTCCGCTTATCACTGGACCT
**Reference gene**		
*GAPDH*	GTCTCCTCTGACTTCAACAGCG	ACCACCCTGTTGCTGTAGCCAA

**Table 2 ijms-23-09271-t002:** The primer sequences of aging-related genes for MeRIP-qPCR.

Primer Name	Forward (5′-3′)	Reverse (5′-3′)
*SIRT3* (3′UTR)	GACCAACATGCTAGAAGTGCGG	AAGCTCGGCATCTGTTGGTTAC
*FOXO1* (3′UTR)	TGGCACACATACCCAGTCTCTG	TGCGCTATGGTGACCTGTTGTA
*E2F3* (3′UTR)	AAACCTTTCTCCTCTGGCCTCC	CAGGGGAGGCAGTAAGTTCACA
*IRS2* (3′UTR)	TGGGAGCAAGGTTGGAGGAAAT	ACACCAAACGGCTTCAGTGAAC
*MST1* (CDS)	CTGTGAGAGCAGTGGGTGATGA	TGGAGTCTGAATGCCTAGCCAG

## Data Availability

All data generated or analyzed in this study are included in this published article.

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
