# Peer review of "Dynamic Alteration Profile and New Role of RNA m6A Methylation in Replicative and H2O2-Induced Premature Senescence of Human Embryonic Lung Fibroblasts"

_ijms, 2022, doi:10.3390/ijms23169271_

Round 1

Reviewer 1 Report

In the present study, the authors showed m6A epitranscription microenvironment in relation to cell senescence, offering new clues for the prevention and intervention of cellular senescence-related diseases. The study is interesting I have some comments provided below

1.     Author should revise their introduction section. It is too long and losing the point of what it should be in many places.

2.     The use of past tense and present tense should be looked at carefully.

3.     The author should list potential future prospect of the study.

4.     In the method section 2.3 please change the last two sentence.

5.     Why Psi and Psp show such big differences must be adequately explained after each result.

6.     The image quality is inadequate, please provide the high-resolution images

7.     Could you explain the scientific rationale behind your choice of a gene list, in 3.5. result “the m6A modification regulated the expression of senescence-related genes”?

Author Response

Reviewer 1:

In the present study, the authors showed m6A epitranscription microenvironment in relation to cell senescence, offering new clues for the prevention and intervention of cellular senescence-related diseases. The study is interesting I have some comments provided below

  1. Author should revise their introduction section. It is too long and losing the point of what it should be in many places.

→ We thank the reviewer very much for the constructive suggestion. Accordingly, we have already modified the introduction section and made it tighter in the new revised manuscript. Now we also show them as below for your check and the relative changes are marked in red color here.

“RNA modification affects the development of many eukaryotic organisms [12] and m6A methylation is the most extensive post-transcriptional modification in RNA, involving RNA nucleation, splicing and stability, etc. [13]. The m6A methylation accounts for 0.1% - 0.4% of adenosine in whole cell RNA and approximately 50% of all methylated ribonucleotides [14], which is associated with many physiological and pathological events related with aging, obesity, immunoregulation, and tumors [15]. The biological functions of RNA m6A are mainly regulated by RNA methyltransferases, demethylases, and binding proteins jointly. The m6A binds to mRNA through RNA methyltransferase, which can be cancelled by demethylase FTO and ALKBH5, as a dynamic and reversible process. The m6A modification can affect the binding of methylated mRNA to proteins and regulate the metabolism of those transcripts directly or indirectly [16].

Cellular senescence related genes have formed a complex network, possibly modified by m6A RNA methylation, which may also interact with oxidative stress [13, 17-19]. Therefore, we will detect the effects of ROS, the dynamic alterations of m6A RNA methylation and the network of target genes during cellular senescence in this study. These characteristics will offer the differences profile between cellular replicative and premature senescence, and provide the potential intervention clues for age-related diseases.”

  1. The use of past tense and present tense should be looked at carefully.

→ We thank the reviewer for the kind reminder and strict request. According to the requirements and instructions, we have altered the corresponding tense thoroughly in the new revision. In general, the regular contents are applied in present tense, and the methods and results are in past tense consistently in the new revision.

  1. The author should list potential future prospect of the study.

→ We thank the reviewer very much for the good suggestion. Accordingly, we have added the potential future prospect in the last paragraph of the discussion section in the new revision. We also show them here in red color as follows for your check.

“From the present study, the RNA m6A methylation modifications open a new window in interpreting the mechanisms for the occurrence and development of senescence and the related diseases. The reversible levels of m6A and the modified proteins will bring about the potential possibility to the prevention and intervention of cell senescence as the targets in the future. The interactions of exogenous oxidative stress, RNA methylation microenvironment and the key genes related with senescence have enriched the epitranscriptomics theory of cell senescence to a better extent.”

  1. In the method section 2.3 please change the last two sentence.

→ We thank the reviewer for the constructive suggestions. We have changed the last two sentences in the method section 2.3 in the revised manuscript as follows.

“Then, the total protein of each group was extracted and the concentration was measured using the BCA method. The total cell ROS level was quantitated relatively using the formula as below.”

  1. Why PSi and PSp show such big differences must be adequately explained after each result.

→ We thank the reviewer for the good suggestion and strict request. Accordingly, we have added the contents to describe the differences between PSi and PSp in the Results section and also supplemented the reasons to explain them in the Discussion part correspondingly in the revised. And we also show them for your check as follows. The relative changes are marked in red color here.

In the Results section, 

On Page 5: 3.1 “The ROS content of PSi was higher than that of PSp. In PSi group with the acute H2O2 treatment, cells accumulated excessive ROS level in a short period. After the following continuous culture for 7 days with normal medium, cells eliminated a certain level of ROS, but they showed persistent and irreversible senescence phenotype with higher content of SA-β-gal in PSp than that in PSi (P < 0.05).”

On Page 7: 3.2 “Therefore, in the premature senescence initiation stage of PSi, the accumulated ROS did not change the m6A level and enzyme activity in rushed time, which of PSi were similar to those of 22PDL. With the development of premature senescence, the RNA methyltransferases activity and m6A level gradually decreased in PSp with an important role.”

On Page 8: 3.3 “PSi and PSp represented the specific stages of premature senescence with differential levels of m6A methylation-related enzymes and binding proteins.”

On Page 14 of the Discussion, we have supplemented the necessary interpretation and added the relative reference as follows here for your check.

“To PSi and PSp, there are some significantly different indicators, including ROS level, SA-β-gal, m6A level and the related enzyme activity respectively, which might be associated with the differential oxidative stress levels and the interactions with epigenetic effects. These results suggest that ROS and RNA methylation play particularly important roles during cellular premature senescence. Another recent study has shown that the modification of RNA m6A can correspond to the damage response caused by ROS [27], which are consistent with our results.”

  1. The image quality is inadequate, please provide the high-resolution images

→ We thank the reviewer very much. Now we have offered the high-resolution images with 500 dpi in the revised figures in the new revision.

  1. Could you explain the scientific rationale behind your choice of a gene list, in 3.5. result “the m6A modification regulated the expression of senescence-related genes”?

→ We thank the reviewer very much. According to the reviewer’s good and helpful suggestion, we supply the reasons to choose the genes related with senescence in the new revised manuscript.

About the choice of the gene list, we selected those genes with key roles in senescence-related pathways, including the longevity regulating pathway (hsa04211), FOXO signaling pathway (hsa04068), cell senescence pathway (hsa04218) and circadian entrainment pathway (hsa04713). Meanwhile, FOXO transcription factors were important determinants in aging and longevity. So, we took FOXO as the core node to select the upstream, downstream and the cross connection genes related to senescence or oxidative stress. We tried our best to explore the complex genes network among them to offer some new clues to elucidate cell senescence and they would also be candidate targeted genes.

We have also added the relative contents in the Results section 3.5 and in the Discussion part as below for your check. The relative changes are marked in red color here.

On Page 11 in the Results section 3.5,

“And then, we firstly focused on FOXO, the important determinant in senescence, to select the upstream, downstream, and the cross connection genes related to senescence or oxidative stress for observing their expression patterns and intrinsic network relationships. They included SIRT3, FOXO1, MST1, IRS2, E2F3, ADCY9, PRKACB, CREB1 and PER2 respectively. Further, we observed the m6A regulation status of the targeted genes. The expression profile of the nine selected genes above were shown in Fig. 5.”

In the Discussion on Page 15, we added the following contents in the new revision.

“Studies have consistently revealed FOXO (Forkhead box O) transcription factors as important determinants in aging and longevity [23]. Therefore, we took FOXO as the node, selected the signaling pathways related to senescence, and detected the genes levels of its upstream and downstream, including the longevity regulating pathway (hsa04211), FOXO signaling pathway (hsa04068), cellular senescence pathway (hsa04218) and circadian entrainment pathway (hsa04713). They have shown the complex network, interaction with each other, in relation to m6A methylation or oxidative stress. Their roles in longevity determination are attractive and need to be fully elucidated.”

The details are also in the attachment.

Reviewer 2 Report

Review comnents are attached

Author Response

Reviewer 2:

The manuscript describes the dynamic profile of RNA m6A methylation and its associated consequences in replicative and H2O2-induced premature senescence of human embryonic lung fibroblasts. The work was carried out in a competent manner, however there exist a number of concerns associated with the substance of the work and the presentation. A representative list of those are remarked upon below and require due attention by the authors prior to any further consideration.

  1. In the introduction, the authors in an effort to refer to literature on the subject, use past tense to describe experiments, with continuing logical conclusions drawn to relate to the present work remaining in the same tense. That creates problems with comprehension of the messages intended and the prospect of work to be carried out. To that end, the authors should revise the grammatical profile of the introduction with appropriate use of verb tenses so that the reader can discriminate between the past literature and the present formulation of the work being done and presented in the manuscript.

→ We appreciate the reviewer’s good suggestions and have changed the past tense into the present tense in Introduction section accordingly. Meanwhile, we have revised the whole grammatical problems carefully in conformity with the requests of the journal in the new revision, besides the Introduction part. We have used the present tense to introduce the general rules and the past tense to describe our methods and results. Thank you very much.

  1. In the concluding remarks of the introduction, the statement made “Considering that RNA methylation might be involved in cellular senescence induced by oxidative stress [13, 28-30], We have showed here the new and direct roles of dynamic alteration of m6A RNA methylation with the regulated target genes in cellular replicative and premature senescence, providing the potential intervention targets for age-related diseases.” is one long sentence. The reader is lost by the end of the paragraph. It should be split into shorter sentences with integral messages.

→ We thank the reviewer very much for the constructive suggestion. Accordingly, we have modified the sentence in the new revision.

 We also show them here for your check as follows. The relative changes are marked in red color here.

“Cellular senescence related genes have formed a complex network, possibly modified by m6A RNA methylation, which may also interact with oxidative stress [13, 17-19]. Therefore, we will detect the effects of ROS, the dynamic alterations of m6A RNA methylation and the network of targeted genes during cellular senescence in this study. These characteristics will offer the differences profile between cellular replicative and premature senescence, and provide the potential intervention clues for age-related diseases.”

  1. The title of section 3 “3.1. Accumulative oxidative stress induced cellular senescence” should be corrected to read “3.1. Accumulating oxidative stress induces cellular senescence” or something along the same lines.

→ We are grateful to the reviewer for the good suggestion. We have revised all of the related titles accordingly in the revision.

We also show them here for your check as follows in red color.

“3.1. Accumulating oxidative stress induces cellular senescence”

“3.2. Global m6A profile of senescence cells is modulated by oxidative stress”

“3.3 Oxidative stress affects the m6A methylation-modulated system during premature senescence”

“3.4. Cellular ROS level is connected with m6A RNA methylation profile”

“3.5. The m6A modification regulates its targeted genes related to senescence”

  1. In the same section, the statement “The cells nuclei became larger and the particulate matters increased.” is not understood. It should be rephrased so that it makes sense.                                                                                  → We really thank the reviewer very much. We have modified this sentence as follows in the new revision. The relative changes are marked in red color here.

“After the acute treatment by H2O2, the cells gradually exhibited the typical phenotype of premature senescence, with the enlarged nuclei and increased cytoplasmic granules.”

  1. In the same section changes occurring during the experiments implemented should be accompanied by the statistically (in) significant notation, not just the increase of decrease of the magnitude of the ROS levels.

→ We thank the reviewer for the constructive suggestions. Accordingly, we have added the descriptions with significant notation in the whole relative results in the revision.

We also show them here, marked in red color for your check as follows.

On Page 6 in 3.1 section,

“… the differences were statistically significant (P < 0.05)”;

“… so the content of SA-β-gal in PSp was higher than that of PSi (P < 0.05).

  1. In section 3.2, the statement “Moreover, the difference of m6A RNA methylation level in PSp had not statistically significance compared with 49PDL (P > 0.05) as shown in Fig. 2A and 2B.” should be corrected to read “Moreover, the difference of m6A RNA methylation level in PSp was not statistically significant compared to 49PDL (P > 0.05), as shown in Fig. 2A and 2B.”.

→ According to the reviewer’s good and strict suggestion, we have revised the sentence as below in the revision. Thank you very much.

“Moreover, the difference of m6A RNA methylation level in PSp was not statistically significant compared to 49PDL (P > 0.05), as shown in Fig. 2A and 2B.”

  1. In the same section, the statement “Compared with 22PDL, the RNA methyltransferases activity in each other group was decreased.” is not understood. It should be rewritten in a comprehensible fashion. What do the authors mean by “each other group”?

→ We thank the reviewer for the strict request. The “each other group” here means other group except 22PDL, such as 35PDL, 49PDL, PSi and PSp group. Now we have rewritten this sentence and the relative contents clearly as follows in the revision. We also show them here for your check. The relative changes are marked in red color here.

On Page 6 in 3.2 section,

“Compared with that in 22PDL group, the RNA methyltransferases activity was decreased in 35PDL, 49PDL, PSi and PSp group respectively with statistical significance (P < 0.05).”

From Page7 to Page 8 in 3.3 section,

“To the mRNA expression, compared with 22PDL, METTL3 was increased, while METTL16 and WTAP was decreased in 35PDL, 49PDL, PSi and PSp group separately (P < 0.05), …

“Compared with 22PDL, METTL14 and METTL16 increased in 35PDL, 49PDL, PSi and PSp group respectively with statistical significance (P < 0.05).”

  1. Here, too, changes occurring during the use of the enzymes (e.g. methyltransferases) should be accompanied by the appropriate statistical notation (as noted above).

→ We thank the reviewer very much for the strict request. Accordingly, we have added the descriptions of statistical notation as noted above in the whole revision. The relative changes are marked in red color here for your check.

On Page 7 in 3.2 section,

“Compared with that in 22PDL group, the RNA methyltransferases activity was decreased in 35PDL, 49PDL, PSi and PSp group respectively with statistical significance (P < 0.05).

“Compared with 49PDL, the RNA methyltransferases activity was increased in PSi (P < 0.05), while the difference between PSp and 49PDL was not statistically significant (P > 0.05).

According to Fig. 2D, the RNA demethylases activity showed no significant difference among all groups (P > 0.05), … ”

On Page 8 in 3.3 section,

… while METTL16 and WTAP was decreased in 35PDL, 49PDL, PSi and PSp group respectively (P < 0.05), and KIAA1429 increased in PSi and PSp (P < 0.05)” “METTL14 revealed no significant difference among each group (P > 0.05).”

“In addition, the mRNA expressions of the six binding proteins, YTHDC1/2, YTHDF1/3, HNRNPA2B1 and HNRNPC showed no significant difference between 49PDL and PSp group (P > 0.05), with only YTHDF2 decreasing (P < 0.05) from Fig. 3A.

Compared with 22PDL, METTL14 and METTL16 increased in 35PDL, 49PDL, PSi and PSp group respectively with statistical significance, but KIAA1429 decreased (P < 0.05) as shown in Fig. 3B. METTL14, METTL3, WTAP and KIAA1429 in PSp group was higher than that in 49PDL separately, while METTL16 was lower than in 49PDL (P < 0.05).

About the RNA methylation binding proteins, YTHDF3 increased with senescence, while each of the other six proteins decreased (P < 0.05). The expression of YTHDC1 and YTHDF1/2/3 in 49PDL was higher than that of PSp respectively (P < 0.05)”.

Thus, the expression of METTL14, METTL16 and YTHDF3 was increased in senescent cells, while KIAA1429, FTO, ALKBH5, YTHDC1, YTHDC2, YTHDF1, YTHDF2, HNRNPA2B1 and HNRNPC was decreased in senescent cells significantly (P < 0.05). In replicative and premature senescent cells, the expression of METTL14, METTL3, WTAP and KI-AA1429 and FTO in PSp group was higher than that in 49PDL, while the expression of METTL16, ALKBH5, YTHDC1 and YTHDF1/2/3 in PSp was lower than that in 49PDL with statistical significance (P < 0.05).

On Page 11 in 3.5 section,

To the mRNA expression, compared with 22PDL, MST1 increased in both senescent cells, while other seven genes were decreased in senescence cells (P < 0.05), and PRKACB had no significant difference (P > 0.05). Compared with 49PDL, IRS2 was increased in PSp, and SIRT3, E2F3, ADCY9, PRKACB, CREB1 and PER2 were separately decreased in PSp (P < 0.05) as shown in Fig. 5A.”

To the protein expression, compared with 22PDL, SIRT3 and E2F3 increased in senescence cells, MST1, ADCY9, PRKACB, CREB1 and PER2 decreased in 49PDL and PSp, and IRS2 decreased in 49PDL and increased in PSp (P < 0.05). Compared with 49PDL, IRS2 increased in PSp, and SIRT3, E2F3, ADCY9, PRKACB, CREB1 and PER2 decreased in PSp (P < 0.05). FOXO1 was no significant difference in each group (P > 0.05) from Fig. 5B”

From Fig. 6B, the m6A methylation modification abundance of SIRT3 and E2F3 in PSp was higher than those in 22PDL and 49PDL (P < 0.05). However, the m6A modification abundance of IRS2 in 49PDL was higher than that in PSp (P < 0.05). The other two genes did not differ statistically between the two groups of senescent cells (P > 0.05).”

  1. In section 3.4, the statement “And then, to the effects of ROS on m6A regulation system, we analyzed the internal correlations between ROS and each RNA methyltransferase, demethylase and methylated binding protein accordingly.” is not understood. Do the authors mean “Subsequently, for the effects of ROS on the m6A regulation system, we analyzed the internal correlations between ROS and each RNA methyltransferase, demethylase, and methylated binding protein accordingly.”

→ We thank the reviewer very much. Yes, the show the correct meaning. We have revised the sentence accordingly as below in the revision.

Subsequently, for the effects of ROS on the m6A regulation system, we analyzed the internal correlations between ROS and each RNA methyltransferase, demethylase, and methylated binding protein accordingly.”

  1. In the title of section 3.5, the use of present tense is better. To that end the suggested change is the following: “3.5. The m6A modification regulates its targeted genes related to senescence”, if that is what the authors intend to convey as a message.

→ We thank the reviewer very much for the accurate expression. Accordingly, we have revised the sentence in the revision. Thank you very much again.

“3.5. The m6A modification regulates its targeted genes related to senescence”

  1. In the discussion section, the statement “When HEFs in young and mid-aged group, the cells had high proliferation ability in a vigorous style.” is missing a verb in the first component sentence. Appropriate corrections should apply.

→ We are grateful with the reviewer’s strict request. Now we have revised the sentence as follows for your check.

When HEFs were in young and mid-aged group, the cells had high proliferation ability in a vigorous style.”

  1. In the same section toward the end, the statement “Therefore, it might be because other members of the FOXO family playing roles in senescence.” should be corrected to read “Therefore, it might be because other members of the FOXO family play roles in senescence.”

→ We are deeply grateful to the reviewer. We have amended the sentence accordingly in the revision.

Therefore, it might be because other members of the FOXO family play roles in senescence.”

  1. In the final paragraph of the text, the statement “In all, the cellular endogenous and exogenous ROS would induce cellular senescence via m6A modification, with specific and differential profile of m6A-related enzymes, binding proteins and the targeted genes.” should be corrected to read colloquially “Collectively, the cellular endogenous and exogenous ROS would induce cellular senescence via m6A modification, with specific and differential profile of m6A-related enzymes, binding proteins, and the targeted genes.”.

→ We appreciate the reviewer’s good help and excellent attitude very much. We now have already revised the sentence accordingly in the revised manuscript.

Collectively, the cellular endogenous and exogenous ROS would induce cellular senescence via m6A modification, with specific and differential profile of m6A-related enzymes, binding proteins, and the targeted genes.”

  1. The references in the reference sections should be modify to comply with the format of the journal.

→ We sincerely thank the reviewer very much. Now we have already modified all the references to comply with the format of the journal carefully. Please check them in the revision.

We hope this re-revised manuscript improved a lot upon the editor and the reviewers’ good guidance and help. Thank you very much again.

The detailed content is also in the attachment.

Round 2

Reviewer 1 Report

No more comments

Author Response

We thank the reviewer very much for the constructive suggestion and kind help. Thank you indeed again.

Reviewer 2 Report

Review comments are attached

Author Response

Dear Prof. Editor,

Thank you very much for inviting us to re-revise our manuscript (ID: 1834816) entitled “Dynamic alteration profile and new role of RNA m6A methylation in replicative and H2O2-induced premature senescence of human embryonic lung fibroblasts” that we would like to be considered for publication in International Journal of Molecular Sciences, Special Issue "Recent Advances in Epigenetics".

We really appreciate the editor’s and the reviewers’ advisable comments and try our best to revise the manuscript accordingly. Based on the detailed and good suggestions, we have checked and revised the grammatical problems word for word again of the whole manuscript. We have also invited an English teacher to help us revise the language. In the text, all of the reference numbers have been placed in square brackets [ ], placed before the punctuation, and provided as superscripts in the main text of the manuscript accordingly.

A point-by-point response to Reviewers’ comments follows this cover letter. Thanks once again for your kind consideration of our manuscript.

Please let us know if there is anything else that needs to be modified, and we really appreciate your help and strict requests. We hope that the corrections will meet your standards. Thank you very much!

Best wishes!

Sincerely yours,

Fan Wu, M.D., Ph.D.

Bo Zhang, M.D., Ph.D

Wenjuan Zhang, M.D., Ph.D.

Reviewer 2:

The manuscript has been revised as per instructions provided. Due attention, however, should be called upon the authors in their effort to improve the presentation of their work in the manuscript.

Specifically, the authors made revisions in the introduction of the manuscript yet their contention that the linguistic picture of the manuscript has improved is incorrect. There are numerous mistakes that render statements incomprehensible or fragmented. To that end, the following examples are shown to remind the authors that they should make careful revisions and not just offer claims about them.

→ We thank the reviewer very much for the strict requirement and good suggestion. Now we have re-read the full text carefully, revised and optimized the relative sentences of the new revision.

  A) In the introduction, the statement “Ginsenoside Rb1 reduce H2O2-induced senescence of human umbilical vein endothelial cells by stimulating sirtuin-1/AMP-activated protein kinase pathway [11]. But the internal mechanisms of ROS inducing senescence are still needed to be clarified further and the epitranscriptome regulation of premature senescence is still in its infancy” should be re-written to read “Ginsenoside Rb1 reduces H2O2 -induced senescence of human umbilical vein endothelial cells by stimulating the sirtuin-1/AMP-activated protein kinase pathway [11]. The internal mechanisms of ROS inducing senescence, however, require further clarification, with the epitranscriptome regulation of premature senescence being still in its infancy”.

→ We thank the reviewer very much for the excellent suggestion. Accordingly, we have re-written the sentences in the revised manuscript using “Track Changes” function as required. We also show them in red for your check as below.

 “Ginsenoside Rb1 reduces H2O2 -induced senescence of human umbilical vein endothelial cells by stimulating the sirtuin-1/AMP-activated protein kinase pathway [11]. The internal mechanisms of ROS inducing senescence, however, require further clarification, with the epitranscriptome regulation of premature senescence being still in its infancy”.

B) At the end of the introduction, the statement “These characteristics will offer the differences profile between cellular replicative and premature senescence, and provide the potential intervention clues for age-related diseases.” should be corrected to read “These characteristics will provide profile differences between cellular replicative and premature senescence, and potential intervention clues for age-related diseases.”. In other words, the revisions made by the authors create new problems still hampering efforts to comprehend the provided statements.

→According to the Reviewer’s constructive suggestion, we have revised the sentence carefully in the new revision by “Track Changes” function. We also offer them here in red for your check expediently. Thank you very much.

“These characteristics will provide profile differences between cellular replicative and premature senescence, and potential intervention clues for age-related diseases.”

  1. In section 3.1, the statement “After the acute treatment by H2O2, the cells gradually exhibited the typical phenotype of premature senescence, with the enlarged nuclei and increased cytoplasmic granules.”, which was supposed to be revised (comment 4 of the original review), should be corrected to read “Following acute treatment with H2O2, the cells gradually exhibited the typical phenotype of premature senescence, involving enlarged nuclei and increased cytoplasmic granules.”.

→We are very grateful to the reviewer for the good guidance. Accordingly, we have corrected the sentence in the revised manuscript using the “Track Changes” function. Now we also show them here marked in the red for your check conveniently.

“Following acute treatment with H2O2, the cells gradually exhibited the typical phenotype of premature senescence, involving enlarged nuclei and increased cytoplasmic granules.”

  1. In section 3.2, the supposedly revised statement “Compared with that in 22PDL group, the RNA methyltransferases activity was decreased in 35PDL, 49PDL, PSi and PSp group respectively with statistical significance (P < 0.05) .” should be corrected to read “Compared with that in the 22PDL group, RNA methyltransferase activity decreased significantly in the 35PDL, 49PDL, PSi, and PSp group (P < 0.05) .”.

→We are very grateful to the reviewer for the good guidance. We have corrected this sentence and other similar sentences in the new revised manuscript as required. We also show them here in red for your check conveniently.

“Compared with that in the 22PDL group, RNA methyltransferase activity decreased significantly in the 35PDL, 49PDL, PSi, and PSp group (P < 0.05).”

About other similar sentences, we have already revised them as below.

On the page 6:

“Compared with that in the 49PDL group, RNA methyltransferase activity increased in the PSi group (P < 0.05), while it was not statistically significant in the PSp and 49PDL group (P > 0.05).”

On the page 7:

“To the mRNA expression, compared with that in the 22PDL group, METTL3 increased, while METTL16 and WTAP decreased in the 35PDL, 49PDL, PSi, and PSp group significantly (P < 0.05), and KIAA1429 increased in the PSi and PSp group (P < 0.05). Compared with that in the 49PDL group, the level of METTL3, WTAP and KIAA1429 increased in the PSp group, respectively (P < 0.05). METTL14 revealed no significant difference in each group (P > 0.05).”

On the page 8:

“To the protein expression, compared with that in the 22PDL group, METTL14 and METTL16 increased significantly in the 35PDL, 49PDL, PSi, and PSp group, ……”

“Thus, the expressions of METTL14, METTL16 and YTHDF3 increased in senescent cells, while they decreased in KIAA1429, FTO, ALKBH5, YTHDC1, YTHDC2, YTHDF1, YTHDF2, HNRNPA2B1 and HNRNPC, respectively”

On the page 11:

“To the mRNA expression, compared with that in the 22PDL group, the level of MST1 increased in either senescent group, while other seven genes decreased significantly (P < 0.05), PRKACB had no significant difference (P > 0.05). Compared with that in the 49PDL group, IRS2 increased in the PSp group, and SIRT3, E2F3, ADCY9, PRKACB, CREB1, and PER2 decreased separately in the PSp group (P < 0.05), as shown in Fig. 5A.”

  1. An important point, remarked upon in the original review process, is that the authors should comply with the directions of references listed in the manuscript. To that end, the authors made corrections on the format of the references in the reference section. The authors should pay attention to the provision of references in the body of the manuscript. To that end, they should comply with the direction of the journal stating that “In the text, reference numbers should be placed in square brackets [ ], and placed before the punctuation; for example [1], [1–3] or [1,3].”. Therefore, references are not provided as superscripts in the main text of the manuscript.

→We are very grateful to the reviewer for the kind help and strict requirement. We have checked every reference number and ensured the reference numbers in square brackets [ ], before the punctuation, and as superscripts in the main text of the revised manuscript using “Track Changes” function. Now we also show the relative changes here in red for your check conveniently.

On the page 2:

“RNA modification affects the development of many eukaryotic organisms and the m6A methylation is the most extensive post-transcriptional modification, involving RNA nucleation, splicing and stability, etc. [12, 13].”

On the page 14:

“ROS signaling plays an essential role in epigenetic processes such as DNA methylation and histone modification from the studies in vitro and in vivo [25, 26]”.

“METTL16 is the RNA methyltransferase, existing as a monomer and has different recognition mechanisms for different RNA substrates, related to the shearing of precursor mRNA [28-30].

Thank you once again for your good guidance and patient help. We hope that the new revisions can meet your strict standards and requirements. We have learnt more from the strictness and the seriousness. Thank you again!

This manuscript is a resubmission of an earlier submission. The following is a list of the peer review reports and author responses from that submission.